# Macrophage-Targeted Dextran Sulfate-Dexamethasone Conjugate Micelles for Effective Treatment of Rheumatoid Arthritis

**DOI:** 10.3390/molecules28020591

**Published:** 2023-01-06

**Authors:** Jiangfan Han, Ren Na, Ningning Zhao, Xiaofeng Yuan, Linke Fu, Jianmei Jing, Airong Qian, Weiliang Ye

**Affiliations:** 1Lab for Bone Metabolism, Key Lab for Space Biosciences and Biotechnology, School of Life Sciences, Northwestern Polytechnical University, Xi’an 710072, China; 2Department of Pharmaceutics, School of Pharmacy, Fourth Military Medical University, Xi’an 710032, China; 3Department of Epidemiology and Health Statistics, Faculty of Military Preventive Medicine, Fourth Military Medical University, Xi’an 710032, China; 4Department of Radiology, Xijing Hospital, Fourth Military Medical University, Xi’an 710038, China

**Keywords:** dextran sulfate, dexamethasone, rheumatoid arthritis, micelles, targeted therapy

## Abstract

Rheumatoid arthritis (RA) is a chronic, systemic immune disease that causes joint affection and even disability. Activated macrophages play an important role in the pathogenesis and progression of RA by producing pro-inflammatory factors. The use of dexamethasone (DXM) is effective in relieving the intractable pain and inflammatory progression of RA. However, long-term use of DXM is strongly associated with increased rates of diabetes, osteoporosis, bone fractures, and mortality, which hinders its clinical use. In this study, the dextran sulfate-cisaconitic anhydride-dexamethasone (DXM@DS-cad-DXM) micelles were prepared to treat RA by selectively recognizing scavenger receptor (SR) on the activated macrophages. The potent targeting property of DXM@DS-cad-DXM micelles to SR was by fluorescence microscope. Additionally, the effective accumulation and powerful anti-inflammatory activity of DXM@DS-cad-DXM micelles were observed in the inflamed joints of adjuvant-induced arthritis (AIA) rats after intravenous administration. Overall, DXM@DS-cad-DXM micelles are a potentially effective nanomedicine for targeted therapy of RA.

## 1. Introduction

Rheumatoid arthritis (RA) is an autoimmune disease mainly characterized by synovial hyperplasia, synovial inflammation, and joint injury [1,2]. Over 60 million cases of RA are estimated to be diagnosed each year worldwide, and many of them suffer multiple systemic complications, even physical disabilities [3,4]. Research has revealed that macrophage-induced inflammation plays an important role in the occurrence and development of RA [5,6,7]. This is due to the activated macrophages releasing a large number of inflammatory cytokines and promoting the differentiation into osteoclasts, ultimately prolonging the inflammatory phase of RA and aggravating the affected joints [8]. The activated macrophages secrete a variety of inflammatory cytokines such as tumor necrosis factor-α (TNF-α), interleukin-1β (IL-1β), and interleukin-6 (IL-6). These cytokines are involved in the development of inflammation, over-activation of the immune system, and metabolism of cartilage and bone [9]. Accordingly, accumulating evidence has suggested that the activated macrophages may be suitable candidate cells for RA-targeted therapy.

In clinical practice, dexamethasone (DXM) is widely used in the treatment of various inflammatory diseases, and it is also the first-line drug for the treatment of RA [10]. Although the pain and other symptoms in RA patients can be relieved by DXM, the associated side effects significantly hinder its clinical application. The small molecule DXM is quickly cleared from the site of inflammation. Therefore, large doses and frequent administration are required clinically to achieve satisfactory therapeutic effects. However, long-term use of DXM is associated with numerous side effects, such as hyperglycemia, osteoporosis, and osteonecrosis [11,12]. Therefore, how to selectively deliver the DXM to the site of RA and how to prolong the action time and decrease the adverse reaction of DXM has become a research hotspot. At present, some delivery vehicles have improved some properties of DXM, whereas this kind of delivery vehicle still has no significant targeting characteristics and the adverse reactions are still relatively serious. Recently, a block copolymer modified by dextran sulfate (DS) has been prepared according to the published report [13]. In vitro and in vivo studies have shown that the copolymer has the ability to target the activated macrophages due to the ligand of DS selectively binds to the SR. The nanocarriers also accumulate efficiently and selectively in the joints of inflamed mice by active targeting strategies [14,15]. Considering the high expression of SR on the surface of activated macrophages and its crucial role in the occurrence and development of RA, we assumed that DS-modified micelles might be a promising carrier for the targeted treatment of RA [16,17]. 

Based on the above studies, we designed and prepared an acid-sensitive amphiphilic copolymer, dextran sulfate-cis-aconitic acid-dexamethasone (DS-cad-DXM), which could be self-assembled into micelles and encapsulate DXM for targeted and efficient treatment of RA (Figure 1). DXM@DS-cad-DXM micelles showed high selectivity to the activated macrophages via SR receptors and rapidly released the DXM at a low pH value, thereby powerfully inhibiting the secretion of pro-inflammatory cytokines. In addition, the therapeutic effect of DXM@DS-cad-DXM micelles on AIA rats was investigated. The results showed that DXM@DS-cad-DXM micelles abundantly accumulate in the inflammatory joints and exert specific anti-inflammatory effects on the activated macrophages, thereby improving the therapeutic effect of RA.

## 2. Results and Discussion

### 2.1. Characterizations of the Conjugates 

As shown in Appendix A, both DEX-cad-DXM and DS-cad-DXM were synthesized by a relatively simple two-step condensation reaction. As depicted in ^1^H NMR spectroscopy (Figure 1a), the signals of 0.75 ppm (a), 0.82 ppm (b), 1.45 ppm (c), 1.49 ppm (d), 2.13 ppm (e), 2.38 ppm (f), 3.48 ppm (j), 3.62 ppm (k), 4.13 ppm (g), and 6.26 ppm (h) were the characteristic proton peaks of DXM. The signals of 4.68 ppm (i) and 5.29 ppm (l) were the characteristic proton peaks of DEX. The signal of 3.38 ppm (m) was the characteristic proton peak of DS. The FT-IR spectrum of DEX-cad-DXM and DS-cad-DXM is shown in Figure 1b; the peak at 3200~3400 and 1004 corresponded to DEX, the peak at 3200~3400,1230 and 980 corresponded to the DS, The peaks at 1705,1662, 1620, 912, and 892 were corresponding to the DXM. These above features suggested the successful synthesis of both DEX-cad-DXM and DS-cad-DXM. The DXM contents of DEX-cad-DXM and DS-cad-DXM were calculated to be 3.4 and 3.6% by HPLC, respectively. 

### 2.2. Preparation and Characterization of Micelles

Due to the amphiphilic property of DEX-cad-DXM and DS-cad-DXM, they could be self-assembled into micelles to encapsulate the free DXM in an aqueous solution, which significantly improved the drug loading capacity of the delivery system. As shown in Figure 2a,b, the CMC of DEX-cad-DXM and DS-cad-DXM were 20 μg/mL and 18 μg/mL, respectively, and the extremely low CMC value is favorable for the formation of micelles. As shown in Figure 2c,d, both DXM@DS-cad-DXM micelles, and DXM@DEX-cad-DXM micelles clearly showed generally spherical in shape. The diameters of DXM@DS-cad-DXM micelles and DXM@DEX-cad-DXM micelles were about 89 nm and 82 nm captured by TEM, respectively. In Figure 2e,f, the hydration sizes of DXM@DS-cad-DXM micelles and DXM@DEX-cad-DXM micelles were 147 ± 12.1 and 137 ± 11.3 nm, respectively, tested by DLS. The appropriate diameters of the two micelles ensure that they can accumulate selectively at the site of inflamed tissue via the effect of extravasation through leaky vasculature and subsequent inflammatory cell-mediated sequestration (ELVIS effect) [18]. The ELVIS effect in inflammatory tissues is similar to the enhanced permeability and retention effect (EPR effect) in tumor tissues, which allows the suitable size of nanocarriers selectively cross diseased blood vessels at the site of inflammation, allowing the drugs to accumulate more in the inflamed tissues than in the normal tissues [19,20]. Zeta potential is one of the important indexes to characterize the nanocarriers. As shown in Table 1, both DXM@DS-cad-DXM micelles and DXM@DEX-cad-DXM micelles possessed negative surface charge, and the zeta potential was −17.4 mV and −19.9 mV, respectively. The negative surface charge of the micelles was due to the abundant hydroxyl groups in DS and DEX molecules. The characteristics of high drug loading are essential for efficient drug delivery systems, and the drug loading of DXM@DS-cad-DXM micelles and DXM@DEX-cad-DXM micelles was 24.3 ± 3.7% and 25.4 ± 3.9%, respectively. Additionally, both DXM@DS-cad-DXM micelles and DXM@DEX-cad-DXM micelles had a high degree of stability in PBS for at least 5 days (Appendix A).

When the delivery system is intravenously administered, it first comes into contact with blood cells and inflammation-associated cells [21]. It is therefore essential to understand the safety of the delivery system. The hemolysis test is a common and effective method for hemocompatibility evaluation [22]. As shown in Figure 3a,b, no obvious hemolysis was found when the concentration of DXM@DS-cad-DXM micelles and DXM@DEX-cad-DXM micelles increased, confirming favorable hemocompatibility. The cytotoxicity of DXM@DS-cad-DXM micelles and DXM@DEX-cad-DXM micelles were also investigated at HUVEC cells and RAW 264.7 cells. As shown in Figure 3c,d, almost no cytotoxicity was detected for the DXM@DS-cad-DXM micelles and DXM@DEX-cad-DXM micelles, even though the concentration of micelles increased to 2000 μg/mL, indicating suitable biocompatibility in cell cultures.

### 2.3. Release Profiles and Targeting Ability Analysis 

During the progression of RA, the local acidic microenvironment results from the decreased local blood supply, increased hematoma and inflammation, enhanced anaerobic metabolism, and massive lactic acid accumulation [23,24]. Numerous studies have reported that polymers bridged by aconitic anhydride are stable under weakly basic conditions but break under weakly acidic conditions [25,26]. Thus, the release of DXM from DXM@DS-cad-DXM micelles and DXM@DEX-cad-DXM micelles was determined in PBS buffer at pH 5.0 and pH 7.4 values to mimic the release of DXM in the blood circulation and inflammatory joints, respectively [27]. As shown in Figure 4c, the amount of DXM released from the DXM@DS-cad-DXM micelles and DXM@DEX-cad-DXM micelles was remarkably dependent on pH value. DXM@DS-cad-DXM micelles and DXM@DEX-cad-DXM micelles released DXM faster at pH 5.0 than at pH 7.4. The faster release of DXM in acidic media is due to the fact that the micellar material is bridged by the acid-sensitive molecule cis-aconitic anhydride. It was expected that the DXM@DS-cad-DXM micelles and DXM@DEX-cad-DXM micelles kept basically stable in the blood circulation and then accumulated in inflammatory joints by ELVIS effect of the selective release of DXM. 

The targeting ability of DXM@DS-cad-DXM micelles toward the activated RAW 264.7 cells was studied by the fluorescent microscope. As shown in Figure 4a,b, a stronger red fluorescence intensity was observed when the activated RAW 264.7 cells were incubated with DXM@DS-cad-DXM micelles than incubated with DXM@DEX-cad-DXM micelles. Nevertheless, when the activated RAW 264.7 cells were with DS+DXM@DS-cad-DXM micelles, the fluorescent signal decreased significantly, indicating a reduced uptake of DXM@DS-cad-DXM micelles. Additionally, when DXM@DS-cad-DXM micelles or DXM@DEX-cad-DXM micelles were incubated with the unactivated RAW 264.7 cells, both of them showed weak fluorescence signals. Moreover, the cellular uptake of DXM@DS-cad-DXM micelles and DXM@DEX-cad-DXM micelles was not reduced by DS pretreatment on unactivated RAW 264.7 cells (Appendix A). These results suggest that the SR, which is expressed on the surface of activated RAW 264.7 cells, is involved in the cellular uptake of DXM@DS-cad-DXM micelles and DXM@DEX-cad-DXM micelles. This phenomenon is consistent with the previous reports [13].

### 2.4. Anti-Inflammatory Activity 

The anti-inflammatory effects of DXM@DS-cad-DXM micelles or DXM@DEX-cad-DXM micelles toward the activated RAW 264.7 cells and unactivated RAW 264.7 cells were tested by ELISA assays. It is well-known that inflammatory cytokines, such as TNF-α, IL-1β, and IL-6, play an important role in the pathological process of RA [28,29]. As shown in Figure 5a, the inflammatory cytokines of TNF-α, IL-1β, and IL-6 were significantly elevated in LPS-treated cells compared to non-LPS-treated cells, whereas the LPS-activated RAW 264.7 cells treated with DXM@DS-cad-DXM micelles remarkably reduced the levels of these cytokines. A certain inhibitory was also detected in the DXM@DEX-cad-DXM micelles group, yet the inhibition effect was relatively lower than that of DXM@DS-cad-DXM micelles. The anti-inflammatory effects of DXM@DS-cad-DXM micelles and DXM@DEX-cad-DXM micelles on the unactivated RAW 264.7 cells were also investigated to further observe the selectivity of micelles on RAW 264.7 cells. As shown in Figure 5b, both DXM@DS-cad-DXM micelles and DXM@DEX-cad-DXM micelles reduced the levels of inflammatory cytokines to some extent, but there was no obvious difference between them. The above data further validated that DXM@DS-cad-DXM micelles had powerful inflammatory inhibition, which was due to the DS specifically binding to SR that overexpressed on activated RAW 264.7 cells.

### 2.5. Selective Biodistribution In Vivo

Biodistributions always intuitively reflect the therapeutic effects and adverse reactions of nanoscale drugs [30]. In order to further evaluate the distribution of the micelles in vivo, the DIR labeled DXM@DS-cad-DXM-DIR micelles, DXM@DEX-cad-DXM-DIR micelles, and free DIR were intravenously administrated in the AIA rats, and the fluorescence signals were captured by using the IVIS. As depicted in Figure 6a, stronger fluorescence signals were observed in the inflamed joints of AIA rats after administrating the DXM@DS-cad-DXM-DIR micelles or DXM@DEX-cad-DXM-DIR micelles. In comparison, the fluorescence signals in the inflamed joints of AIA rats were much weaker for the free DIR group. This phenomenon indicates that the prepared micelles passively selectively accumulate into the inflamed joint of the AIA rats due to the “ELVIS” effect [31]. Additionally, the fluorescence signals of the DXM@DS-cad-DXM-DIR micelles group in the inflamed joints showed much higher than that of the DXM@DEX-cad-DXM-DIR micelles group, which was ascribed to the DS inherent high affinity for SR [13]. The quantitative analysis of fluorescence intensity was also carried out according to fluorescent images (Figure 6b). The fluorescence intensity in the inflamed joints for the DXM@DS-cad-DXM-DIR micelles group was significantly higher than that DXM@DEX-cad-DXM-DIR micelles group (~2.8-fold). After 24 h administration, the main organs of the rats were isolated, and the fluorescence distribution was observed. As depicted in Figure 6a,c, the fluorescence intensity decreased significantly in all the main organs, indicating that both the micelles can be quickly removed from the blood circulation. However, there were still strong fluorescence signals in the inflamed joints of rats for the two micelle groups. Furthermore, the fluorescence intensity in the inflamed joints for the DXM@DS-cad-DXM-DIR micelles group was 3.6 times that of the DXM@DEX-cad-DXM-DIR micelles group. The above studies indicated that DXM@DS-cad-DXM micelles achieved good targeting ability and appropriate biodistribution by intravenous administration.

### 2.6. Therapeutic Efficacy of AIA Models 

The hind paw thickness and clinical scores were documented to evaluate the therapeutic effects of the micelles on the AIA rat model. As shown in Figure 7a,c, the model control rats suffered significant swelling in the paw, whereas the swelling was markedly alleviated in the DXM@DS-cad-DXM micelles group and DXM@DEX-cad-DXM micelles group, and the DXM@DS-cad-DXM micelles had the better swelling relief effect. After four doses of treatment, then the paw morphology of the rats in the DXM@DS-cad-DXM micelles group was very close to that of normal rats. The administration of free DXM had little effect on alleviating the swelling of the paws. As shown in Figure 7b, the arthritis score peaked after 12 days of modeling, indicating that the AIA rat model was successfully established. DXM@DS-cad-DXM micelles significantly reduced the clinical scores of AIA rats, and the effect was obviously superior to that of free DXM and DXM@DEX-cad-DXM micelles. DXM@DS-cad-DXM micelles had the most successful anti-rheumatoid arthritis effect, which was attributed to both the passive targeting-ELVIS action and active targeting action by DS actively recognized SR [14]. Recently, self-assembled dextran sulfate (DS) nanoparticles (NPs) have been applied for targeted therapy of osteoarthritis and atherosclerosis, which exhibit enhanced anti-inflammatory efficacy and reduced side effects through specific targeting to the scavenger receptor class A (SR-A) on activated macrophages [32,33]. In contrast, the therapeutic efficacy of DXM@DEX-cad-DXM micelles was relatively lower, as it was only dependent on the passive targeting-ELVIS to improve the anti-rheumatoid efficacy. The levels of TNF-α, IL-1β, and IL-6 in the synovial cavity of the right hind paws were tested by ELISA kits. As shown in Figure 7d, the levels of TNF-α, IL-1β, and IL-6 were largely boosted in the model control groups, whereas the secretion levels of these inflammatory cytokines were significantly reduced in AIA rats given DXM@DS-cad-DXM micelles. Free DXM and DXM@DEX-cad-DXM micelles also reduced the secretion of inflammatory factors, but the inhibitory efficacy was significantly lower than that of DXM@DS-cad-DXM micelles. These results suggested that DXM@DS-cad-DXM alleviates the clinical symptoms by inhibiting the secretion of related inflammatory factors and thereby treating rheumatoid arthritis.

Hematoxylin–eosin was performed to evaluate the histological changes in ankle joints after treatments. As shown in Figure 8a, in the model control rats, the joint cavity was narrowed due to the excessive proliferation of inflammatory cells, and significant erosion of cartilage was present. The treatment of free DXM alleviated the inflammatory infiltrate slightly, but tissue damage was still evident. DXM@DEX-cad-DXM micelles showed strong anti-inflammatory activity, yet obvious synovial hyperplasia and mild articular cartilage erosion can still be observed. In contrast, the inflamed joints in AIA rats treated with DXM@DS-cad-DXM micelles showed almost normal articular cavities with no cartilage erosion and mild inflammatory hyperplasia. The Micro-CT was conducted to investigate the structure of inflamed paws in AIA rats after treatments. As shown in Figure 8b, the number of bone trabeculae and bone mineral density in the model control group was obviously decreased, suggesting serious bone destruction in AIA rats. After treatment with DXM@DS-cad-DXM micelles, the number of bone trabeculae and bone mineral density of AIA model rats were significantly increased. In contrast, both free DXM and DXM@DEX-cad-DXM micelles could not remarkably improve the impaired bone quality in the AIA model rats. These results suggested that DXM@DS-cad-DXM micelles have an inhibitory effect on bone destruction caused by inflammation, which was consistent with the above HE results.

### 2.7. Safety Evaluation In Vivo

Body weight change is one of the important indexes to evaluate the safety of drugs [16,34]. Throughout the course of treatment, the body weight of rats in all treatment groups continued to increase slowly except for the model control rats, which decreased significantly (Figure 9a), indicating that the given formulations did not lead to weight loss. Long-term administration of DXM was associated with hyperglycemia, and the levels of blood glucose were measured. As shown in Figure 9b, the mean blood glucose of the rats in each treatment group was within the normal range and did not fluctuate significantly. ALP, AST, CRE, and BUN are important indicators for reflecting the functions of the liver and kidney [35]. The data showed that all the indexes were within the normal range after the treatment (Figure 9c–f). HE was performed to further evaluate the damage to the liver and kidney (Appendix A). No obvious histopathology abnormality or pathological changes were found in the liver and kidney for each group after the treatment. Together, these results manifested that DXM@DS-cad-DXM micelles can be a prospective clinical treatment for RA.

## 3. Materials and Methods

### 3.1. Materials 

Dextran Sulfate (DS, MW5000 Da), Dextran (DEX, MW5000 Da), Dexamethasone (DXM), Cis-aconitic anhydride (CAD), N-Hydroxysuccinimide (NHS), 4-N,N-dimethylaminopyridine (DMAP), and 1-Ethyl-(3-(dimethylamino)propyl)-carbodiimide hydrochloride (EDC·I) were purchased from J&K Scientifific Ltd. (Beijing, China). Dulbecco’s modified Eagle’s medium (DMEM) and fetal bovine serum (FBS) were purchased from Thermo Fisher Scientifific (China) Co., Ltd. (Shanghai, China). Lipopolysaccharide (LPS) was obtained from Invitrogen Technologies Company (Carlsbad, CA, USA). 

### 3.2. Cell Culture and Animals

HUVEC cells and RAW264.7 cells were purchased from Cell Resource Center, Shanghai Institute of Biological Sciences, Chinese Academy of Sciences (Shanghai, China) and placed in DMEM medium in a cell incubator (37 °C, 5% CO_2_). Male SD rats (weight = 180–200 g) were obtained from Air Force Medical University Animal laboratory center. All procedures are in accordance with the guidelines for the Care and Use of Laboratory animals, which are approved by the Animal Ethics Committee (Xi’an, China). To induce AIA rats, the male SD rats were injected intradermally with 0.05 mL of Freund’s complete adjuvant through the hind footpad of the rats. The successful induction of AIA rats was identified through a significant increment in paw thickness and joint score [36,37]. 

### 3.3. Syntheses of DS-cad-DXM Conjugate 

The synthesis route of DS-cad-DEX conjugate is shown in Appendix A. First, carboxylated DXM (DXM-COOH) was obtained by a reaction between dexamethasone (DXM) and Cis-aconitic anhydride (CAD), using the NHS and DMAP as the catalyst. Next, the DXM-COOH was activated by NHS and EDCI in a dioxane solution for 8 h. Then, the DS was dissolved in H_2_O and dropped into the above reaction mixture. After 24 h of reaction, the solution was dialyzed for 48 h to obtain DS-cad-DXM conjugate. DEX-cad-DXM was prepared as a control conjugate by using Dextran (DEX) to replace DS. The 1H NMR spectra and FT-IR were used to characterize the successful synthesis of DS-cad-DXM and DEX-cad-DXM. 

### 3.4. Preparation and Characterization of DS-cad-DXM Conjugate Micelles (DXM@DS-cad-DXM)

The micelles were prepared by dialysis method [38]. First, the DS-cad-DXM and DXM were dissolved into the DMSO and then slowly added into the deionized water with vigorous stirring. Finally, the solution was dialyzed for 12 h to obtain the micelles of DXM@DS-cad-DXM. The DXM@DEX-cad-DXM micelles were prepared following a similar method and were used as the control. 

The diameter, polydispersity index, and zeta potential of the DXM@DS-cad-DXM micelles and DXM@DEX-cad-DXM micelles were analyzed by using a Particle Analyzer (Delsa Nano C, Beckman Coulter, CA, USA). The morphology of DXM@DS-cad-DXM micelles and DXM@DEX-cad-DXM micelles were determined by using a JEOL JEM-1011 TEM (JEOL-100CXII, Ltd., Tokyo, Japan). The critical micelle concentration (CMC) of DXM@DS-cad-DXM micelles and DXM@DEX-cad-DXM micelles were measured by fluorescence spectroscopy. Furthermore, the stability of the micelles was also studied. 

The DXM content in DXM@DEX-cad-DXM micelles consists of two parts: one part is a chemically linked DXM, and the other part is encapsulated by micelles. A certain number of micelles were dissolved into an HCl solution and shaken for 6 h to produce DXM. Then, the drug loading of the DXM was determined by HPLC (Flexar, Perkinelmer, Shelton, CT, USA).

### 3.5. Release Profiles of DXM from DXM@DS-cad-DXM 

The DXM release property from DXM@DS-cad-DXM micelles was performed in a phosphate buffer solution at pH 5.0 or pH 7.4. In brief, the DXM@DS-cad-DXM micelles were dissolved into the release medium and then transferred to a dialysis bag (MWCO = 1500 Da). Samples were taken out at the designed time intervals, and the amount of DXM was analyzed by HPLC.

### 3.6. Biocompatibility Evaluation of DXM@DS-cad-DXM

Hemolysis characteristics are a key criterion for the biocompatibility of drug delivery systems. The hemocompatibility properties of DXM@DS-cad-DXM micelles and DXM@DEX-cad-DXM micelles were determined by an ultraviolet-visible (UV-Vis) spectrophotometry (UV-1800, Shimadzu, Kyoto, Japan). In brief, the DXM@DS-cad-DXM micelles and DXM@DEX-cad-DXM micelles solutions of various concentrations were added to 100 μL diluted rat blood. Next, the formulations were thoroughly mixed via vortex and incubated for 2 h. Finally, the solutions were isolated, and the OD value was measured by an ultraviolet-visible (UV-Vis) spectrophotometry. 

The MTT assay was executed to further evaluate the biocompatibility of the DXM@DS-cad-DXM micelles and DXM@DEX-cad-DXM micelles. The HUVEC cells and RAW 264.7 cells were seeded into 96-well plates and incubated for 24 h. Then, the cells were treated with various concentrations of the DXM@DS-cad-DXM micelles and DXM@DEX-cad-DXM micelles solutions for 24 h. Finally, the OD value in the well was measured by using a Microplate Reader (Bio-Rad Laboratories, Hercules, CA, USA). 

### 3.7. Targeting Ability of DXM@DS-cad-DXM 

A fluorescent microscope was used to observe the targeting ability of DXM@DS-cad-DXM micelles and DXM@DEX-cad-DXM micelles toward the LPS-activated RAW 264.7 cells and the unactivated RAW 264.7 cells. The micelles were labeled with RhB to form DXM@DS-cad-DXM-RhB micelles and DXM@DEX-cad-DXM-RhB micelles for fluorescence monitoring. In brief, RAW 264.7 cells were planted into a coverslip-containing 24-well plate and incubated for 24 h. Then, the cells were activated by LPS (100 ng/mL) for 12 h. Whereafter, the fresh cell culture medium containing DXM@DS-cad-DXM-RhB micelles or DXM@DEX-cad-DXM-RhB micelles replaced the previous culture medium, followed by incubation for 2 h. Additionally, 5 mg/mL of DS was co-incubated with the RAW 264.7 cells for 30 min before being treated with the micelles as a control. Finally, the cells were stained with DAPI and washed with PBS three times. The cellular uptake of DXM@DS-cad-DXM-RhB micelles and DXM@DEX-cad-DXM-RhB micelles towards the unactivated RAW 264.7 cells was also conducted similarly with the above method. 

### 3.8. Anti-Inflammatory Efficacy of DXM@DS-cad-DXM

The pro-inflammatory cytokines levels of TNF-α, IL-1β, and IL-6 were detected by ELISA kit. In brief, the RAW 264.7 cells were planted into a coverslip-containing 6-well plate and incubated for 24 h. Then, the cells were activated by LPS (100 ng/mL) for 12 h. Whereafter, the fresh cell culture medium containing DXM@DS-cad-DXM micelles and DXM@DEX-cad-DXM micelles replaced the previous culture medium, followed by incubation for 24 h. The anti-inflammatory efficacy of DXM@DS-cad-DXM micelles and DXM@DEX-cad-DXM micelles towards the unactivated RAW 264.7 cells was also conducted similarly with the above method. 

### 3.9. Biodistribution of DXM@DS-cad-DXM 

The biodistribution of DXM@DS-cad-DXM micelles and DXM@DEX-cad-DXM micelles were captured by using the IVIS (Caliper, Hopkington, MA, USA). Briefly, the AIA rats were injected intravenously with DIR labeled DXM@DS-cad-DXM-DIR micelles and DXM@DEX-cad-DXM-DIR micelles solution. The rats were anesthetized after 2 h post-injection, and the imaging of fluorescence on joints was visualized by the IVIS system. In addition, after 24 h administration, the main organs of the rats were isolated, and fluorescence imaging was observed by the IVIS system. The fluorescent signals on the organs were also quantitatively calculated by the Maestro™ 2.4 software (CRi, Woburn, MA, USA).

### 3.10. In Vivo Therapeutic Efficacy of DXM@DS-cad-DXM 

The rats were randomly divided into five groups (n = 5). One group was normal rats; the other four groups were AIA model rats. All five groups were administered different formulations every 5 days for a total of four doses, and the paw thickness and clinical scoring were examined every 2 days. The AIA rats were sacrificed 2 days after the last administration, and the knee joints were collected for histopathological analyses. The secretion levels of TNF-α, IL-1β, and IL-6 in joint tissues were detected by ELISA kit. Additionally, the micro-CT (Munich, Germany) analysis was performed on the inflamed paw of rats to assess whether the bone was damaged. 

### 3.11. Safety Evaluation DXM@DS-cad-DXM

At the end of treatment, the alanine transaminase (ALT), aspartate aminotransferase (AST), blood urea nitrogen (BUN), creatinine (Crea), and blood glucose were measured to evaluate the in vivo biocompatibility of the DXM@DS-cad-DXM micelles and DXM@DEX-cad-DXM micelles.

### 3.12. Statistical Analysis 

The data were analyzed by SPSS13.0 software (SPSS, Chicago, IL, USA). Significant differences were determined by *t*-test or one-way analysis of variance (ANOVA), and *p* < 0.05 was considered statistically significant (GraphPad Prism 8 software, San Diego, CA, USA).

## 4. Conclusions

In this work, the macrophage targeting copolymer DS-cad-DXM and non-targeting copolymer DEX-cad-DXM were synthesized. The two amphiphilic copolymers self-assembled into spherical vehicles to encapsulate the DXM and thus formed DXM@DS-cad-DXM micelles and DXM@DEX-cad-DXM micelles, respectively. Both of them possessed more than 23% of the drug load, and the diameters were about 100 nm. The appropriate particle size of the micelles is beneficial for the passive targeting of inflammatory joints by the ELVIS effect. Additionally, DXM@DS-cad-DXM micelles selectively delivered DXM into the activated macrophages through targeted binding of DS to SR. Thus, compared with the non-targeted DXM@DEX-cad-DXM micelles, DXM@DS-cad-DXM micelles showed a stronger anti-inflammatory activity against activated macrophages, a more selective biodistribution in vivo, and a more powerful anti-rheumatoid activity in AIA model rats. In summary, DXM@DS-cad-DXM micelles have great potential for targeted therapy in rheumatoid arthritis.

## Data Availability

Not applicable.

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
