# Peer review of "Macrophage-Targeted Dextran Sulfate-Dexamethasone Conjugate Micelles for Effective Treatment of Rheumatoid Arthritis"

_molecules, 2023, doi:10.3390/molecules28020591_

Round 1

Reviewer 1 Report

The work by Han et al. explores the utility of dexamethasone delivered in dextran sulfate-cis aconitic acid micelles to treat rheumatoid arthritis by targeting inflammatory macrophages through scavenger receptor. Overall, the study opens the possibility of using dextran mediated drug delivery for therapeutics against rheumatoid arthritis and similar inflammatory diseases.

Major comment:

1.    To claim that the DXM@DS-cad-DXM micelles targeted macrophages through scavenger receptors, it is recommend that experiments be done to knockdown scavenger receptor (Msr1) in macrophages to see if the micelles lose their potency in the knockdown cells. Alternatively, pharmacological inhibitors of scavenger receptors such as fucoidan may also be used to demonstrate this.

Minor comments:

1.    Scheme 1 has a typographical error – “Stable Internal Circulation”. Fig 1 has a typographic error in “wavenumber” on the x-axis twice.

2.    In the abstract and main text, use the expanded form of AIA (Adjuvant-induced arthritis) at the first occurrence. Similarly, expand EPR and ELVIS effects at the first occurrence.

3.    Grammar fixes are required for sentences in lines 38-40, 57-60 and 171-173.

4.    Fig 2 has missing units for graphs 2a and 2b in the x-axis.

5.    Fig 6b seems to be indicative of the 2h timepoint. Please indicate this in the figure legend for clarity.

6.    Scavenger receptors have been implicated in a variety of inflammatory diseases. In the discussion, can you please add on the potential applicability of these micelles in targeting other inflammatory diseases?

Reviewer 2 Report

Good Day,

The manuscript describes interest results, sustained by many modern analytical techniques, in an area of great interest nowadays. The introduction can be improved by adding few more references from the last decade. The English trough the whole manuscript requires only small moderate changes. The discussion and the conclusions were developed based on the obtained results.

After performing small corrections and adding more citations relevant to the fields of arthritis reumatoid treatment and of micellar solutions, the manuscript can be accepted for publication.

Best regards!
